# Exemplar-free Continual Representation Learning with Symmetric Distillation

## Abstract

Continual learning strives to train a model in a sequential manner by learning from new tasks while retaining information about old tasks. Treating this as a common classification problem leads to catastrophic forgetting, especially in deep learning settings, where knowledge of old tasks is forgotten as soon as a model is optimized on new tasks. Existing solutions tackle this problem by imposing strict assumptions, such as the availability of exemplars from previously seen classes or a warm start of a model on many classes before starting the continual learning. While effective on known benchmarks, such assumptions can be impractical and do not directly address the stability-plasticity dilemma in continual learning. In this paper, we follow a recent push in the field to tackle continual learning in the exemplar-free cold-start setting. We propose *Model-in-the-Middle (MITM)*. The idea behind MITM is to separate the learning of new classes and retention of past class knowledge by using two distinct models. We propose a learner with symmetric distillation from both models, enabling us to learn evolving representations as new tasks arrive. We show that explicitly separating and balancing old and new tasks through symmetric distillation helps absorb large distribution shifts, mitigating the stability gap. Our approach is simple yet outperforms the state-of-the-art in the challenging exemplar-free cold-start continual learning setting.

## 1 Introduction

Continual representation learning addresses a fundamental limitation in deep learning: when a neural network is trained on a new task, it immediately forgets most information about old tasks (French, 1999). This catastrophic forgetting results from the stability-plasticity trade-off, as identified by Grossberg (2012). An overly stable model may fail to learn new tasks, while an overly plastic model risks forgetting previous tasks. Thus, the central challenge in continual learning is to strike the right balance between stability and plasticity, allowing models to learn representations for both old and new tasks (De Lange et al., 2021; Wang et al., 2024).

Continual learning methods approach the *stability-plasticity* dilemma from various angles. Some focus on *stability*; Starting from a strong initialization point and attempting to change the model as little as possible for the new classes (Zhang et al., 2023; McDonnell et al., 2024). Dynamic network approaches, for instance, expand the model's architecture as new data emerges, based on the assumption that model capacity is limited and adapting to new features will overwrite old ones, leading to forgetting (Rusu et al., 2016; Yan et al., 2021). For models with fixed or limited capacity, regularization-based approaches control plasticity by either adjusting the optimization direction (data regularization) (Li & Hoiem, 2017; Buzzega et al., 2020) or evaluating the importance of each parameter and regularizing the parameters (Kirkpatrick et al., 2017; Liu et al., 2020).

Another group of methods emphasizes *plasticity*, enabling the model to adapt to new patterns while remembering old patterns (Wu et al., 2019; Buzzega et al., 2020). Exemplar-based methods aim to preserve the knowledge of previous tasks by replaying exemplars stored in memory (Chaudhry et al., 2019; Rebuffi et al., 2017) or by generating synthetic examples using generative models (Smith et al., 2021). On the other hand, prototype-based approaches use exemplars to prevent semantic drift by re-calculating class prototypes after each stage (Rebuffi et al., 2017; De Lange & Tuytelaars, 2021; Zhu et al., 2021). Model rectification methods assume that catastrophic forgetting will occur, and they attempt to mitigate it by correcting the biases introduced (Wu et al., 2019). Similarly, prototype-based

approaches that do not use exemplars or pre-trained models correct the biases by drift estimation and feature consolidation (Gomez-Villa et al., 2024; Magistri et al., 2024). Knowledge distillation techniques transfer information from an expert model, trained on previous tasks, to a student model that is learning the current task (Li & Hoiem, 2017; Buzzega et al., 2020).

The need for exemplars remains central in most methods to avoid bias toward recent tasks. Storing previous-task samples, however, has multiple practical limitations, such as memory constraints, legal concerns, and dealing with sensitive and private data that should be safeguarded (Venkatesan et al., 2017). Moreover, exemplar-based solutions do not generalize to vision-language domains, as keeping exemplars of all possible textual descriptions is not tractable. This has prompted a shift towards exemplar-free continual learning (Wang et al., 2024), where many methods use prototypes to replace exemplars. A downside of exemplar-free prototype-based methods is that they are prone to semantic feature drift (Yu et al., 2020). To mitigate the drift, these methods work under warm-start settings, beginning with a large first task or a pretrained model (Zhu et al., 2021; Petit et al., 2023) to start with good representations that stay unchanged. Recent works advocate studying exemplar-free cold-start continual learning (Magistri et al., 2024; Gomez-Villa et al., 2024), as this setup provides the most challenging setting for continual learning. In the absence of exemplars and warm-start, models employ complex techniques to correct the bias (Gomez-Villa et al., 2024) or consolidate features (Magistri et al., 2024).

The goal of this paper is to perform continual learning without the need for exemplars, prototypes, warm starts, or additional corrections. We strive for a simple solution that stays close to a canonical supervised objective, without the need to balance information from old and new tasks through careful hyperparameter tuning. To that end, we introduce *Model-in-the-Middle (MITM)*. MITM balances plasticity and stability by separating the learning of new representations from the integration of past and current knowledge. Given a network to be optimized, we take a leading, middle, and trailing version. The trailing model retains past knowledge, while the leading model learns new information. The middle model distills knowledge from both, achieving a natural balance between stability and plasticity, effectively mitigating task-recency bias. To the best of our knowledge, this is the first exemplar-free cold-start method that addresses task-recency biases without relying on complex correction or consolidation mechanisms. Empirical results show that MITM provides state-of-the-art results in the exemplar-free cold-start setting. Supporting analysis shows that our approach comes without a significant bias toward initial or most recent tasks while mitigating forgetting.

## 2  RELATED WORK

**Warm-start methods**  In the exemplar-free setting, most methods use regularization to prevent forgetting. LwF (Li & Hoiem, 2017) applies regularization to class probabilities, while EWC (Kirkpatrick et al., 2017) applies it to model weights. Others approach regularization by estimating parameter importance (Aljundi et al., 2017; Zenke et al., 2017). However, these methods often suffer from significant semantic drift (Yu et al., 2020), leading to further research aimed at reducing this issue. FeTrIL (Petit et al., 2023) addresses this by generating pseudo-features based on the difference between old and new prototypes, helping the model adapt to new classes. Panos et al. (2023) freeze the feature extractor after the first task, using pseudo-features from the initial task to handle subsequent ones. In FeCAM (Goswami et al., 2024), the authors leverage the heterogeneous feature distribution in class-incremental learning, using an anisotropic Mahalanobis distance instead of the standard Euclidean metric for better classification. These methods are typically evaluated in warm-start scenarios, which begin with a large initial task or pretrained models. Starting from strong representations allows these models to focus more on stability by freezing the feature extractor (Petit et al., 2023; Panos et al., 2023; Goswami et al., 2024), limiting plasticity or relying on pre-trained models (Zhang et al., 2023; McDonnell et al., 2024). Recently, the shortcomings of warm-start settings have sparked renewed interest in exemplar-free cold-start class-incremental learning (Magistri et al., 2024; Gomez-Villa et al., 2024).

**Correction-based methods**  Works addressing the difficult problem of exemplar-free cold-start continual learning must manage the effects of increased plasticity. High plasticity can cause feature drift, leading to a bias toward the most recent tasks, as the model adapts quickly but risks losing information from earlier tasks (Masana et al., 2022). In earlier approaches, this bias was corrected by using exemplars (Wu et al., 2019) or applying cosine normalization (Hou et al., 2019). In the

exemplar-free setting, methods such as Zhu et al. (2021; 2022) mitigate this bias by employing prototypes, which approximate class means in the feature space and help calibrate the classifier. Prototypes are also used to model semantic drift, as in Gomez-Villa et al. (2024), and can even eliminate the need for a classification layer. Direct prevention of semantic drift is addressed in Magistri et al. (2024) by consolidating features and balancing them through prototype rehearsal. While prototype-based methods are promising, they require assumptions about class distributions, and the subsequent parameters must be carefully estimated for effective use.

**Disentangling past and present task representations** Another way to balance stability-plasticity without using exemplars is by separating what is relevant for past and present tasks. One group of methods (Mittal et al., 2021; Ahn et al., 2021) addresses this separation by identifying the Softmax operation as a major cause of catastrophic forgetting. They show that applying Softmax separately to the previous and current task classes can mitigate this issue. This separated Softmax also helps reduce the stability gap (De Lange et al., 2023), as demonstrated in Caccia et al. (2022). On one side, dynamic networks explicitly separate the past and present knowledge by expanding the model's backbone, or neurons, as new data emerges (Rusu et al., 2016; Yan et al., 2021), based on the assumption that the model's capacity is limited in representing both old and new tasks effectively. However, these methods require expandable memory budgets, making them unsuitable for incremental learning on edge devices. On the other side, distillation-based methods, such as Javed & Shafait (2019); Buzzega et al. (2020); Mittal et al. (2021), compress past and present knowledge into a single student model by using a previous copy of the model as a teacher. Our approach strikes a balance between these extremes, where we use three separate models, each an expert in the current task, past tasks, and a combination of the two. The idea of using three separate models for exemplar-free incremental learning is also explored in Lee et al. (2019); Zhang et al. (2020). In contrast, we jointly train all models on the current task data, preventing abrupt shifts in target distributions.

## 3 METHOD

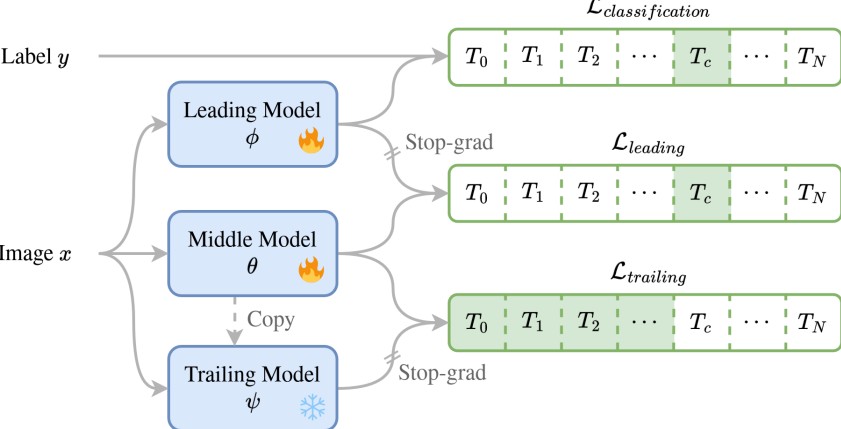

Figure 1: **Overview of Model-in-the-Middle (MITM)** with a visual summary showing which task logits each loss is applied to. Every image-label pair is propagated through all three identical models. The leading model is trained exclusively on the classes in the current task $T_c$, which separates learning new representations from merging previously learned ones. Symmetric distillation of the leading and the trailing logits naturally balances the representations in the middle model. At each task boundary, the middle model's parameters are copied to the trailing model, preserving the newly acquired knowledge. No gradients are propagated through connections marked as stop-grad. The snowflake symbol indicates frozen network parameters.

### 3.1 MODEL-IN-THE-MIDDLE

In class-incremental representation learning, we are given data that is split up into $N$ tasks, denoted as $\mathcal{T} = \{T_1, ..., T_c, ..., T_N\}$., with $T_c \in \mathcal{T}$ indicating the current task. The dataset $\mathcal{D}$ is split into an

equivalent number of subsets $\mathcal{D}_{T_c} \subset \mathcal{D}$ and a held-out evaluation set $\mathcal{E}_{T_c} \subset \mathcal{E}$. We are sequentially given access to these tasks. A task consists of all samples and labels $(x, y) \sim \mathcal{D}_{T_c}$. Classes in a task $\mathcal{C}_{T_c} \subset \mathcal{C}$ are mutually exclusive with all other tasks, i.e. $\mathcal{C}_{T_i} \cap \mathcal{C}_{T_j} = \emptyset$. The samples drawn from these tasks conform to the i.i.d. assumption. This contrasts the dataset $\mathcal{D}$ as a whole, as it presents the tasks sequentially over time, thereby undermining the stationarity assumption. In this work, we investigate the cold-start exemplar-free setting, where all tasks are of equal size, and we do not allow the storage of exemplars. With these definitions, the objective is to find parameters $\theta$ of the function $f(\cdot; \theta)$ to minimize

$$\min_{\theta} \mathcal{L} = \mathbb{E}_{(x,y) \sim \mathcal{D}} \left[ \ell \left( f(x; \theta), y \right) \right] \tag{1}$$

where $\ell(\cdot, \cdot)$ denotes the softmax cross-entropy loss function. This is problematic when approximating the expectation, as $\mathcal{D}$ is not a stationary distribution. We, therefore, sequentially minimize the stationary tasks

$$\min_{\theta} \sum_{i=1}^{N} \mathcal{L}_{T_i}. \tag{2}$$

The current task must be optimized not to affect the model's predictions of past tasks. Finding a suitable balance between retaining the past tasks and adapting to the current ones is a crucial challenge. We propose a new approach to this problem, Model-in-the-Middle (MITM), which obtains this balance by using symmetric knowledge distillation. At its core, we hypothesize that *using the same mechanism for stability and plasticity simplifies the search for a balance between the two*. Our approach is visualized in Figure 1. Formally, we introduce

$$\mathcal{L}_{T_i} = \mathcal{L}_{trailing} + \mathcal{L}_{leading} + \mathcal{L}_{classification}. \tag{3}$$

Our approach uses two identical models in addition to $\theta$ but with separate parameterizations. We denote the parameters of the leading model as $\phi$ and the trailing model as $\psi$. The leading model's function is to learn to classify the samples of the current task, similarly to Caccia et al. (2022). The trailing model retains knowledge of past tasks, and it is obtained by copying the middle model's parameter when it is fully trained. Finally, the middle model is tuned to classify all tasks.

The classification loss term is used to optimize the leading model. A standard softmax cross-entropy function $\ell$ is applied only to the logits of the classes in the current task. We denote the logits of the classes in task $T_c$ with $f_{T_c}(x; \phi)$:

$$\mathcal{L}_{classification} = \mathbb{E}_{(x,y) \sim \mathcal{D}_{T_c}} \left[ \ell \left( f_{T_c}(x; \phi), y \right) \right]. \tag{4}$$

The leading model provides a key benefit to our approach. As this model is not concerned with past tasks, we can apply an unmodified supervised learning approach, ensuring we do not compromise its ability to learn new representations with regularization or stability terms.

The middle model $\theta$ combines two knowledge distillation losses. The distillation from the leading model $\phi$ provides plasticity, while the trailing model $\psi$ provides stability. For both of these terms we use mean squared error to match the target logits, shown to be effective in Buzzega et al. (2020). As the leading model is only trained on the classes in the current task, we only distill on the corresponding $n$ logits in task $T_c$, indexed using $i$. Additionally, note the leading model is only optimized through its respective loss in Eq. 4, as we do not propagate the gradients from this loss to $\phi$.

$$\mathcal{L}_{leading} = \mathbb{E}_{(x,y) \sim D_{T_c}} \left[ \frac{1}{n} \sum_{i=1}^{n} (f_{T_c}(x; \theta)^i - f_{T_c}(x; \phi)^i)^2 \right] \tag{5}$$

where $i$ is used to iterate over the subset of logits. Similarly, we only distill the logits of the trailing model for the classes in which it has been trained. These logits from all past tasks up to $T_c$, are given by $f_{T_{0:c-1}}(x; \phi)$. We do not propagate gradients through the trailing model, and the parameters of the trailing model are, therefore, only modified upon model copy after the task. We allow the trailing model to update its batch norm statistics to better accommodate distribution shifts between tasks, as described in Szatkowski et al. (2023). Finally, this term does not contribute to the first task as there are no past logits.

$$\mathcal{L}_{trailing} = \mathbb{E}_{(x,y) \sim D_{T_c}} \left[ \frac{1}{n} \sum_{i=1}^{n} (f_{T_{0;c-1}}(x;\theta)^i - f_{T_{0;c-1}}(x;\psi)^i)^2 \right] \tag{6}$$

Our motivation for choosing logit distillation builds on the observations on feature distillation in Magistri et al. (2024), who argue that feature distillation avoids feature drift, a key factor in forgetting, but does so by constraining plasticity in all directions. In contrast, by distilling based on past logits, we constrain only the feature subspace relevant to these logits and allow for plasticity in the underlying feature representations. This is crucial, as this allows the underlying features to recombine to accommodate the new classes while retaining logit activation's on past classes.

In summary, the leading model allows for a few simplifications compared to other approaches. Due to the separation of learning new representations from merging existing ones, we allow for unconstrained optimization of the current task. Additionally, there is no need to explicitly balance this plasticity, as the symmetric distillation naturally introduces this balance. The lack of hyperparameters for balancing and additional codebooks, memory banks, or pretraining makes for a simple yet effective approach.

## 4 EXPERIMENTS

### 4.1 TRAINING SETUP

We train a slimmed ResNet18 model based on Lopez-Paz & Ranzato (2017), as is commonly used in continual learning literature (Buzzega et al., 2020; Zhu et al., 2021; Magistri et al., 2024). Following Buzzega et al. (2020), we apply random crops and horizontal flips to both the input stream and buffer exemplars. All introduced methods are trained for 100 epochs using SGD with a momentum of 0.9 and no weight decay. The learning rate is set to 0.01 and is reduced by a factor of 0.1 at epochs 50 and 85. The order of classes is fixed to ensure reproducibility.

Our experiments are conducted on S-CIFAR-100 (Krizhevsky et al., 2009; Rebuffi et al., 2017), which consists of 10 incremental steps, each containing 10 classes, in a cold-start setting following Magistri et al. (2024); Gomez-Villa et al. (2024). We also perform experiments on TinyImageNet (Wu et al., 2017) under the same cold-start configuration and with 10 steps. Unlike the warm-start setting, where the model is pretrained on a portion of the data (typically half of the tasks) before incremental learning, the cold-start setting is more challenging as models must learn incrementally from scratch.

### 4.2 EVALUATION METRICS

To provide insight into the performance of the method both during training and the resulting model, we report multiple metrics. We report Final Average Accuracy (**FAA**), the accuracy averaged over all tasks after training on the final task. Additionally, we report Average Incremental Accuracy (**AIA**) as in Magistri et al. (2024), which averages the evaluation accuracy after every task, providing some insight into the performance *during* training. As in De Lange et al. (2023), we report average minimum accuracy (**min-ACC**) and average forgetting (**Forgetting**) to provide insight into how well knowledge is retained. Specifically, to compute the average minimum accuracy, we evaluate the entire test set after every training iteration and retain the per-task minimum observed value, as specified in De Lange et al. (2023). This provides insight into the transient forgetting, known as the stability gap, which is hypothesized to contribute to overall forgetting.

We introduce a new metric Final Accuracy standard deviation (**FA$\sigma$**) to provide further insight into task bias, as many methods exhibit a bias towards the most recent task (Masana et al., 2022). To this end, we report the standard deviation of the per-task final accuracy. The intuition is that a method with significant bias towards any task has a larger variance in final task accuracies than an unbiased method with equal accuracy for all tasks. FA$\sigma$ is not independent of FAA, and it should always be considered alongside it.

For every task $T_k$ we define a training dataset $\mathcal{D}_{T_k}$ and a held-out evaluation set $\mathcal{E}_{T_k}$. We define the accuracy $\mathbf{A}(\mathcal{E}_{T_k}, f(\theta_{T_k})) \in [0, 1]$ as the percentage of correct top-1 classifications on evaluation set $\mathcal{E}_{T_k}$ with the parameters $\theta_{T_k}$ just after training on task $T_k$. The task bias metric is defined as:

Table 1: Comparison to the state-of-the-art on **S-CIFAR-100 exemplar-free cold-start 10-step** setting. We observe MITM outperforming other methods in final average accuracy, average incremental accuracy, and forgetting, indicating MITM is better able to learn new knowledge incrementally without significantly degrading past knowledge. This is further supported by MITM's exceptionally low task bias score, indicated with FA$\sigma$, which approaches that of the lower bound. Additionally, the high average minimum accuracy shows we successfully avoid the stability gap. [†]Reproduced using mammoth (Buzzega et al., 2020). [¶]Reproduced using author's implementation. [‡]Excerpted from Gomez-Villa et al. (2024). [§]Estimated from author's results. - indicates results are not available. Mean and standard deviation of three runs is provided.

|  | FAA↑ | AIA↑ | Forgetting↓ | min-ACC↑ | FA$\sigma$↓ |
|---|---|---|---|---|---|
| Joint training (upper bound) | 71.78±0.5 |  |  |  | 2.94±0.3 |
| EWC (Kirkpatrick et al., 2017)[†] | 8.18±0.5 | 24.86±0.1 | 83.86±0.8 | 0.0±0 | 22.13±1.2 |
| LwF (Li & Hoiem, 2017)[†] | 9.66±0.1 | 26.33±0.1 | 89.17±0.5 | 0.0±0 | 27.92±0.1 |
| DMC (Zhang et al., 2020) | 37[§] | - | - | - | - |
| FeTrIL (Petit et al., 2023)[‡] | 34.94±0.5 | 51.20±1.1 | - | - | - |
| EFC (Magistri et al., 2024)[¶] | 44.20±0.5 | 57.88±0.8 | 25.26±0.7 | 37.17±0.4 | 13.53±1.2 |
| LwF+LDC (Gomez-Villa et al., 2024)[‡] | 45.4±2.8 | 59.5±3.9 | - | - | - |
| MITM | **49.57±0.5** | **61.52±0.3** | **14.58±0.2** | **44.00±0.7** | **3.91±0.3** |

$$\mathbf{FA}\sigma = \sqrt{\frac{1}{N}\sum_{i=1}^{N}(\mathbf{A}(\mathcal{E}_{T_i}, f(\theta)) - \mu)^2} \text{ where } \mu = \frac{1}{N}\sum_{i=1}^{N}\mathbf{A}(\mathcal{E}_{T_i}, f(\theta)) \tag{7}$$

## 4.3 COMPARITIVE ANALYSIS

In our first experiment, we compare MITM to baselines and state-of-the-art exemplar-free approaches in the cold-start setting. Additionally, we consider forgetting and task bias metrics to provide insight. In Table 1, MITM outperforms other methods by a margin in S-CIFAR100, while Table 2 shows the same improvements for S-TinyImageNet. We observe that FeTrIL, tailored for the warm-start setting, struggles in the cold-start setting. This is also the case for other warm-start methods such as Zhu et al. (2021); Goswami et al. (2024); Zhu et al. (2022) as previously highlighted by Magistri et al. (2024). EFC and LwF+LDC perform better in the cold-start setting and represent the state-of-the-art. MITM outperforms these methods in both accuracy metrics. In terms of final average accuracy we improve by more than 9% over LwF+LDC and EFC, while reducing forgetting by 42% compared to the latter, indicating we do not prevent forgetting by simply not learning. The average minimum accuracy observed for past tasks is close to the final average accuracy, indicating we do not suffer from the transient forgetting observed in De Lange et al. (2023). This benefit in minimum accuracy is also observed for EFC, which also employs a separated softmax.

Mitigating the stability gap prevents potential degradation of past task performance, contributing to preventing a bias to certain tasks, and allowing EFC to improve over the baselines in terms of task bias. MITM further improves in this metric, obtaining a result close to joint training performance. This indicates it does not compromise in task bias to inflate final average accuracy, which can be the case when biased to the most recent task. We attribute this additional balance shown by MITM to the symmetrical distillations losses. On the more difficult S-TinyImageNet dataset, we still observe some task bias compared to the upper bound, although we obtain an improvement over EFC. We also include Deep Model Consolidation (DMC) introduced in Zhang et al. (2020), which shares an equivalent architecture to our method. In contrast, our method optimizes all three models jointly on the current task data. This is deliberate, as asynchronously updating the models introduces a sudden shift in distillation targets. Distributional shifts, such as a task boundary, can introduce the stability gap and lead to the degraded accuracy observed. Finally, we provide some additional exemplar-based results in appendix A.2. In the appendix, we show how to extend our approach to work with

Table 2: Comparison to the state-of-the-art on **S-TinyImageNet exemplar-free cold-start** 10-step setting. We observe MITM to outperform other methods in terms of accuracy and forgetting. [‡]Excerpted from Magistri et al. (2024). [¶]Reproduced using author's implementation.

| | FAA↑ | AIA↑ | Forgetting↓ | FA$\sigma$↓ |
|---|---|---|---|---|
| Joint training (upper bound) | 59.58±0.3 | | | 3.43±0.2 |
| EWC (Kirkpatrick et al., 2017)[‡] | 8.00±0.3 | 15.70±0.4 | - | - |
| LwF (Li & Hoiem, 2017)[‡] | 26.09±1.3 | 45.14±0.9 | - | - |
| FeTrIL (Petit et al., 2023)[‡] | 30.97±0.9 | 45.60±1.7 | - | - |
| EFC (Magistri et al., 2024)[¶] | 33.83±1.1 | 47.59±0.77 | 25.45±2.8 | 11.01±1.3 |
| LwF+LDCGomez-Villa et al. (2024) | 34.2±0.7 | 46.8±1.1 | - | - |
| MITM | **42.18±0.6** | **55.14±0.7** | **23.29±0.7** | **8.29±0.3** |

Table 3: Comparison to the state-of-the-art on **S-CIFAR-100 exemplar-free cold-start 20-step** setting. MITM slightly outperforms EFC, while achieving significantly less forgetting and task bias. [‡]Excerpted from Magistri et al. (2024). [¶]Reproduced using author's implementation.

| | FAA↑ | AIA↑ | Forgetting↓ | FA$\sigma$↓ |
|---|---|---|---|---|
| Joint training (upper bound) | 71.78±0.5 | | | 2.94±0.3 |
| EWC (Kirkpatrick et al., 2017)[‡] | 17.7±2.4 | 31.02±1.2 | - | - |
| LwF (Li & Hoiem, 2017)[‡] | 17.44±073 | 38.39±1.1 | - | - |
| FeTrIL (Petit et al., 2023)[‡] | 23.28±1.2 | 38.8±1.1 | - | - |
| EFC (Magistri et al., 2024)[¶] | 31.87±0.9 | 47.22±0.72 | 24.68±0.8 | 14.24±0.7 |
| MITM | **32.54±0.1** | **48.04±0.4** | **15.21±0.9** | **9.43±0.7** |

exemplars and report results in low memory settings, where our approach is more effective than existing solutions.

As in Magistri et al. (2024), we provide an evaluation of MITM in the cold-start 20 step setting of S-CIFAR100 in Table 3. We use the same hyperparameters, with the exception of the learning rate which is reduced to 0.005. In this setting, it is more difficult to prevent forgetting of past tasks due to the increased number of steps and reduced numer of samples per step. Both EFC and MITM observe degraded performance under these constraints, especially in terms of task bias, although MITM retains the advantage in all metrics.

Table 4: **Ablation on disentanglement of old and new tasks** on S-CIFAR-100. Logit distillation is a similar method to LwF. Extending this with a separated softmax shows a clear benefit to the average minimum accuracy, indicating this mitigates the stability gap. MITM is obtained by introducing a leading model to logit distillation. This naturally separates the softmax, while the symmetric distillation significantly reduces task bias. We provide more details for these baselines in appendix A.1.

| | FAA↑ | AIA↑ | Forgetting↓ | min-ACC↑ | FA$\sigma$↓ |
|---|---|---|---|---|---|
| Logit distillation | 9.87±0.3 | 26.90±0.5 | 83.26±0.7 | 0.0±0.0 | 26.70±0.4 |
| Logit distillation + separated softmax | 39.50±0.6 | 53.02±0.5 | 13.81±0.6 | 32.29±1.0 | 12.47±0.3 |
| MITM | **49.57±0.5** | **61.52±0.3** | **14.58±0.2** | **44.00±0.7** | **3.91±0.3** |

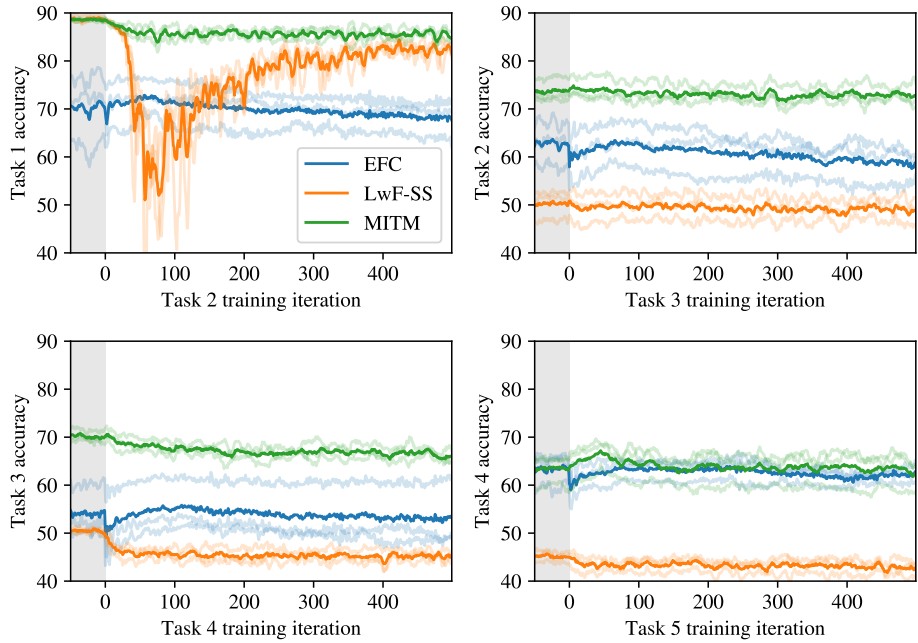

Figure 2: **Stability gap:** Comparison of MITM to EFC, and LwF with separated softmax. The validation accuracy for task N is shown during the initial training steps on task N+1. We show three runs and highlight the mean. For LwF-SS, detailed in appendix A.1, we can observe significant transient forgetting, i.e. a stability gap, for the first task. On other tasks, the stability gap is mitigated. In EFC, we observe a subtle stability gap, indicating it is significantly mitigated. MITM does not yield a similar gap.

In summary, we show MITM to obtain state-of-the-art results in CIFAR100 and TinyImageNet. This result is achieved with very little task bias, as indicated by our introduced metric, outperforming prototype based methods such as EFC.

## 4.4 ANALYSIS OF MODEL-IN-THE-MIDDLE

Below, we outline multiple ablation studies to better understand our approach.

**Disentangling old and new tasks.** In Table 4 we compare MITM's disentanglement strategy with other ablated variants. Logit distillation, equivalent to LwF, suffers strongly from excess forgetting due to the global softmax. Separating this significantly improves the method and avoids the stability gap. This method, LwF-SS, still suffers from a significant task-recency bias, as shown by the obtained FA$\sigma$. The addition of a leading model allows us to naturally balance stability and plasticity, and we see MITM outperforms others in the task bias metric.

**Stability gap analysis.** In Figure 2, we show the evaluation accuracy of the previous task for every training iteration of the current task, as specified in De Lange et al. (2023). In this work, the authors observe most methods to exhibit significant forgetting of the previous task, which is subsequently recovered to a certain extent. This effect happens in the initial training iterations of the subsequent task. Similar to Caccia et al. (2022), we observe applying a separated softmax mitigates this issue on most tasks, which is the case for all methods shown. In the case of LwF+SS, we still observe the stability gap for the first task. For EFC, we can observe the stability gap occurring on a small scale, indicating it is largely mitigated. We observe no forgetting of previous tasks at the task boundary for MITM. We attribute this to the the addition of the leading model. As this leading model is updated through many small training steps, it avoids sudden distributional changes and slowly propagates these to the middle model.

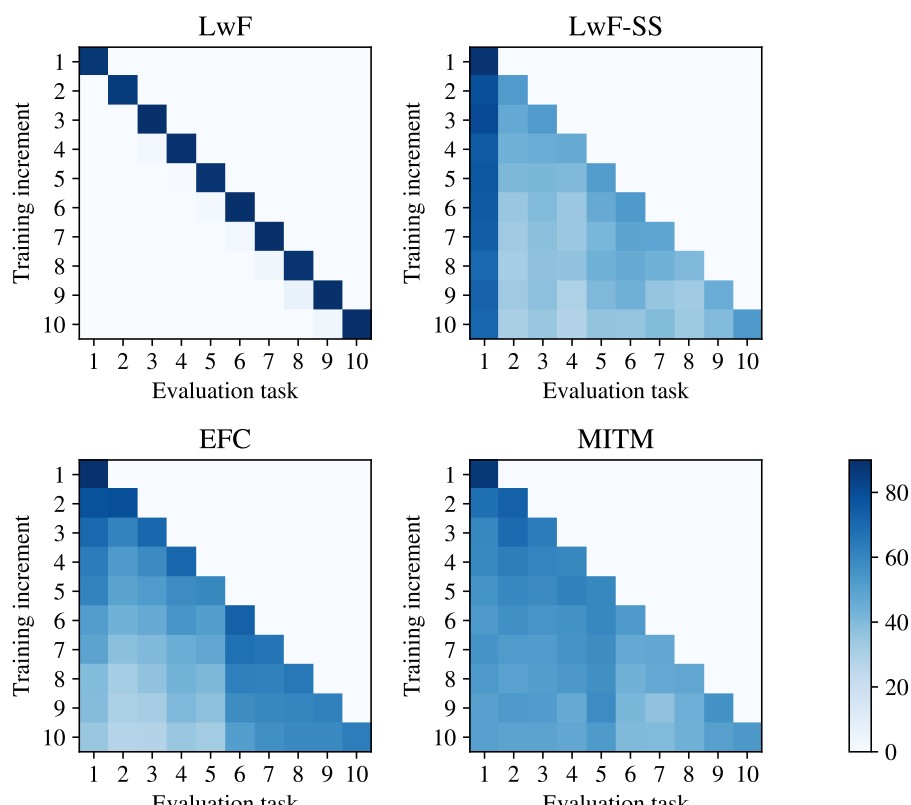

Figure 3: **Per-task accuracy after every task on S-CIFAR100.** LwF observes excess plasticity, leading to catastrophic forgetting. Applying a separated softmax (LwF-SS) significantly reduces this, but is biased towards the initial and most recent tasks. EFC lessens the bias towards the initial task, but still suffers from task-recency bias. MITM does not observe a clear bias towards any task.

**Task-recency bias.** In Figure 3, we show the accuracy on all tasks after each trained task. For LwF, we observe the catastrophic forgetting to significantly bias the method towards the most recent tasks. This forgetting is alleviated with LwF-SS, but a strong bias towards the initial and most recent tasks is still observed. In EFC, a slight bias towards both the initial and most recent tasks is observed, as we can observe some forgetting the tasks directly after the initial task. This indicates that the prototypes used to balance the classifier may not be sufficient. The symmetric distillation does not bias the middle model towards any particular task, naturally introducing this balance without necessitating exemplars or prototypes.

**Stability-plasticity trade-off.** We further investigate the balance between the leading and trailing distillation by introducing a balancing hyperparameter $\lambda$. That is, we update Equation 3 to be

$$\mathcal{L}_{T_i} = \mathcal{L}_{trailing} + \lambda \mathcal{L}_{leading} + \mathcal{L}_{classification} \tag{8}$$

In Table 5 we provide results for various choices for $\lambda$. We can observe choosing a lower $\lambda$, which biases towards the trailing loss, slightly increases performance. As initial task is learned to a higher accuracy, a bias towards these slightly increases accuracy. Pushing this bias further, with $\lambda = 0.5$, we obtain the lowest amount of forgetting, which is generally highest on the initial tasks. However, we observe this bias towards the earlier tasks is at the cost of the accuracy on later tasks, as any choice other than $\lambda = 1.0$ increases the task bias metric. This shows symmetric distillation is effective at balancing past and current tasks.

Table 5: **Ablation on balancing leading and trailing model** on cold-start S-CIFAR100 with 10 steps. We observe MITM to be robust to different choices for lambda, with $\lambda = 1$ providing the least task bias. The mean and standard deviation of three runs are given.

| Lambda | FAA↑ | AIA↑ | Forgetting↓ | FA$\sigma$↓ |
|--------|------|------|-------------|-------------|
| 0.5 | $49.82_{\pm0.4}$ | $61.34_{\pm0.2}$ | $\mathbf{11.79_{\pm0.1}}$ | $4.63_{\pm0.4}$ |
| 0.8 | $\mathbf{50.21_{\pm0.4}}$ | $\mathbf{62.34_{\pm0.5}}$ | $13.29_{\pm0.5}$ | $4.51_{\pm0.6}$ |
| 1.0 | $49.57_{\pm0.5}$ | $61.52_{\pm0.3}$ | $14.58_{\pm0.2}$ | $\mathbf{3.91_{\pm0.3}}$ |
| 1.2 | $48.88_{\pm0.3}$ | $61.45_{\pm0.2}$ | $15.77_{\pm0.6}$ | $5.12_{\pm0.7}$ |
| 2.0 | $47.18_{\pm0.5}$ | $61.22_{\pm0.5}$ | $19.80_{\pm1.3}$ | $6.02_{\pm0.6}$ |

## 5 CONCLUSION

In this paper, we introduce MITM, a simple symmetric distillation method for continual representation learning that closely resembles canonical supervised learning. Our method does not depend on exemplars, warm start, or additional bias correction. We motivate the separation of learning new representations and merging existing ones, and show that this approach naturally mitigates the stability gap and reduces task bias. Building on these properties, our approach obtains state-of-the-art results for exemplar-free cold-start class-incremental learning. Additionally, we introduce a new metric for task bias and investigate MITM's ability to balance past and current tasks. Finally, we perform ablations on the components of our method regarding the disentanglement strategy and the stability-plasticity balance to highlight their contribution.

We employ logit distillation to allow underlying features to adapt, providing increased plasticity over feature distillation. It is possible that insights from regularization methods can further refine the allowed plasticity to avoid forgetting. This paper focuses on offline continual learning without exemplars and in low-memory settings. Extending our method to be complementary to exemplar-based settings and online continual learning scenarios might be exciting directions to investigate.

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

# A APPENDIX

## A.1 IMPLEMENTATION DETAILS OF ABLATION METHODS

In Table 4 we introduce logit distillation. This method closely resembles LwF (Li & Hoiem, 2017), and its objective is given by

$$\mathcal{L} = \mathbb{E}_{(x,y) \sim \mathcal{D}_{T_c}} \left[ \ell \left( f(x; \phi), y \right) + \frac{1}{n} \sum_{i=1}^{n} (f_{T_{0;c-1}}(x; \theta)^i - f_{T_{0;c-1}}(x; \psi)^i)^2 \right] \tag{9}$$

where $\ell(\cdot, \cdot)$ denotes the softmax cross-entropy loss function. This method can be extended with a separated softmax by simply restricting the logits on which the softmax operates to the current task

$$\mathcal{L} = \mathbb{E}_{(x,y) \sim \mathcal{D}_{T_c}} \left[ \ell \left( f_{T_c}(x; \phi), y \right) + \frac{1}{n} \sum_{i=1}^{n} (f_{T_{0;c-1}}(x; \theta)^i - f_{T_{0;c-1}}(x; \psi)^i)^2 \right] \tag{10}$$

## A.2 EXEMPLAR-BASED EXTENSION

While our approach is designed for exemplar-free setting, it is flexible in nature and able to generalize to the exemplar-based setting with few modifications. To make use exemplars from a small buffer $\mathcal{M}$, we modify the expectation of equations in leading loss (Eq. 5) and trailing loss (Eq. 6) to be

$$\mathbb{E}_{(x,y)\sim D_{T_c}\cup\mathcal{M}}. \tag{11}$$

As in Chaudhry et al. (2019); Buzzega et al. (2020); Wu et al. (2019), we approximation the expectation by sampling half the batch from $D_{T_c}$ and the other half $\mathcal{M}$, as to avoid class imbalance from the replay buffer. Additionally, we modify the total loss (Eq. 3) to add a cross entropy loss on the labels for the middle model. To avoid introducing the stability gap, we scale the contribution of the cross entropy loss by the fraction of epochs that have passed for the current task, $\eta \in [0,1]$. This way, when large distributional changes are occur at the task boundary, $\eta$ is (close to) zero.

$$\mathcal{L}_{middle} = \eta \cdot \mathbb{E}_{(x,y)\sim D_{T_c}\cup\mathcal{M}} \left[\ell(f_{T_c}(x;\theta), y)\right] \tag{12}$$

We provide a comparison to exemplar-based metods in Table 6. All methods have been run using hyperparemeters specified in Boschini et al. (2022). It should be noted for fair comparison, as argued in Zhou et al. (2023), that the compared methods only make use of two model copies, while MITM uses three. However, this additional model copy is not used to provide additional storage of past tasks.

In exemplar based setting, we see MITM is competitive in the low buffer regime. In Table 6 we can observe these generally strong exemplar based methods to under-perform due to the low number of exemplars, especially in the case with only 20 per class. In the case of DER and DER++, these methods do not employ a separated softmax, and therefore also obtain a low minimum accuracy, indicating significant transient forgetting. This is alleviated in X-DER, which combines the logit distillation from DER together with separated softmax from ER-ACE to obtain state-of-the-art results in the cold-start setting.

We note that both of these methods, ER-ACE and X-DER, observe a significant reduction in minimum accuracy when the number of exemplars is small. This indicates these methods are not fully able to mitigate the transient forgetting in this case, possibly contributing to a reduced final average accuracy.

Table 6: **S-CIFAR-100 exemplar-based cold-start comparison in the 10-step setting.** MITM performs comparably in the low memory buffer regime. In ER-ACE and X-DER the use of a separated softmax significantly improves the average minimum accuracy, indicating the stability gap is reduced. This is further improved by MITM using symmetric distillation. Improving exemplar-based results is an interesting avenue for future work.

| | | FAA↑ | AIA↑ | Forgetting↓ | min-ACC↑ | FA$\sigma$↓ |
|---|---|---|---|---|---|---|
| $\mathcal{M} = 500$ | DER (Buzzega et al., 2020) | 33.61±1.6 | 54.93±0.7 | 52.85±1.7 | 1.86±0.3 | 15.47±1.0 |
| | DER++ (Buzzega et al., 2020) | 37.53±0.3 | 55.49±0.7 | 51.51±0.3 | 4.49±1.8 | 15.58±1.1 |
| | ER-ACE (Caccia et al., 2022) | 38.12±1.3 | 54.34±0.7 | 38.80±0.8 | 25.46±1.2 | **9.91±1.6** |
| | X-DER (Boschini et al., 2022) | 48.13±0.5 | 61.54±0.5 | **23.72±0.1** | 27.71±2.2 | 12.19±0.5 |
| | MITM | **50.16±0.9** | **63.34±0.5** | 27.93±1.6 | **43.19±0.9** | 10.63±0.4 |
| $\mathcal{M} = 200$ | DER (Buzzega et al., 2020) | 23.26±0.9 | 43.49±1.3 | 70.65±1.9 | 1.82±0.9 | 22.19±0.3 |
| | DER++ (Buzzega et al., 2020) | 26.68±0.7 | 45.55±1.2 | 65.10±0.9 | 1.59±0.4 | 19.26±0.2 |
| | ER-ACE (Caccia et al., 2022) | 29.39±0.9 | 48.35±0.8 | 45.98±1.7 | 18.96±0.4 | **10.76±1.0** |
| | X-DER (Boschini et al., 2022) | 35.83±0.5 | 54.61±0.4 | 41.94±0.6 | 19.76±1.8 | 20.89±0.1 |
| | MITM | **46.86±0.4** | **61.53±0.1** | **31.73±0.8** | **39.36±0.2** | 13.52±0.1 |

We note that compared to its exemplar-free variant, MITM with 200 exemplars has a reduced accuracy and significantly increased task bias. This result is in contrast to Chaudhry et al. (2019), which

argue the benefit of these tiny memories. Improving the effectiveness of exemplars in our framework is an interesting direction for future work.

