# OpenReview forum: "Exemplar-free Continual Representation Learning with Symmetric Distillation"
_ICLR.cc/2025/Conference — ICLR 2025 Conference Withdrawn Submission_

### Official Review · Reviewer_GTZZ · 2024-10-16

**Soundness:** 2
**Presentation:** 3
**Contribution:** 3
**Rating:** 5
**Confidence:** 5

**Summary:**

The authors address the problem of exemplar-free class-incremental learning in the challenging cold-start setting, where classes are equally distributed among tasks, and no initial training is performed on a large first task. This contrasts with the less challenging warm-start setting, where the backbone exploits a well-trained representation before incremental learning.

The authors propose the Model-in-the-Middle (MITM) approach, which aims to separate the learning of new classes from the retention of past class knowledge. Specifically, for each task, two models are trained (leading and middle models), and a frozen model from the previous task (trailing model) is used for distillation. The leading model is trained with a cross-entropy loss applied only to the logits of the current task's classes, without any regularization term, and aims to learn the feature representation for the current task. The middle model is trained with a symmetric distillation loss, which consists of two logit distillation losses applying a mean square error to match target logits: the first logit distillation term matches the logits of the middle model on the current task's classes with the logits of the leading model for the same classes; the second logit distillation term matches the logits of the middle model on the previous task's classes with the trailing model, which is a frozen copy of the middle model before training.

At the end of training, the symmetric distillation loss balances the middle model's representations, enabling classification across all tasks. The authors evaluate the proposed approach using multiple metrics. Beyond standard metrics in continual learning (CL)—such as final average accuracy, average incremental accuracy, forgetting, and minimum accuracy—the authors introduce a new metric called Final Accuracy Standard Deviation to provide insights into task-recency bias. The proposed method demonstrates state-of-the-art performance on S-CIFAR-100 (10-step, 20-step) and S-Tiny-ImageNet (10-step) across the four metrics, without relying on any prototype reharsal approach, which is a common strategy in the literature.

Regarding their analysis, the authors show that employing separate softmax layers for current and previous task classes mitigates the stability gap in the offline CL setup, confirming the results found by Caccia et al. (2022) for online CL. They also ablate the proposed approach against logit distillation with separated softmax layers. Finally, they discuss the stability-plasticity trade-off, highlighting that the method is robust with respect to hyperparameter selection in the leading model's cross-entropy loss.

**Strengths:**

- The paper is well-written. The authors effectively contextualize the problem they aim to solve, starting with the introduction and continuing through the related work section. The methodology section clearly outlines the challenges and how they are addressed. I appreciate Figure 1, which, in its simplicity, helps the reader follow the methodology easily. The experimental section is also clearly presented.

- The proposed method, Model-in-the-middle (MITM), introduces a simple and effective mechanism to reduce task recency bias and improve average accuracy by employing the proposed symmetric logit distillation loss on the middle model and cross entropy loss on the leading one, as demonstrated in Table 4 of the ablation study.

- Compared to previous exemplar-free approaches, the authors' original contribution lies in training the leading model to learn the current task classes, while the middle model balances the representation without requiring any prototypes. This approach avoids assumptions about the feature distribution that are typically necessary for prototype rehearsal, as well as the need for strategies to compensate for drift.

- I find the proposed metric for task-recency bias (Final Accuracy Standard Deviation) particularly interesting, as it enables a better analysis of the impact of task-recency bias on final performance, alongside the analysis of stability gap in the offline incremental setting.

**Weaknesses:**

**Methodology Weaknesses**

Overall, I believe that the methodology requires additional theoretical or empirical investigation to further strengthen the novel contributions of the paper. Below, I summarize my major concerns:

- While the use of separate softmax layers and logit distillation on past logits adds value to the approach, these techniques are not entirely novel. Similar strategies have already been explored in offline exemplar-based methods such as X-DER[a], as well as in online incremental learning  Caccia et al[b]. Additionally, the concept of distillation using both past and current logits is well-established in X-DER, though in the context of methods with exemplars rather than exemplar-free settings. The current approach demonstrates that logit distillation improves performance even in exemplar-free scenarios, but it remains unclear why it is effective in this context. A more thorough empirical investigation into why logit distillation works under such conditions would greatly enhance the contribution of the paper, which otherwise remains limited to the introduction of the MITM model.

- The authors mention that they allow the trailing model to update the batch normalization statistics to account for distribution shift across tasks, but no empirical evidence of the resulting performance improvement is provided. Are the batch normalization statistics also updated for the middle and leading models?

**Experiments Weaknesses**

Overall, I believe the experimental section is weak, and there are several ambiguities that make it difficult to fully understand the performance of the proposed method. Additionally, the authors do not address the limitations of their approach. Below, I summarize my major concerns:

- The authors mention that they train a slimmed ResNet-18 model for their experiments (Lopez-Paz & Ranzato, 2017 [c]). This model has three times fewer feature maps across all layers compared to the full ResNet-18, resulting in approximately 1 million parameters and feature space size equals to 160 (compared to about 11 million parameters and feature space size equals to 512 in the full version). To the best of my knowledge, all competitors (FeTriL[d], EFC[e], LwF+LDC[f] ) use the full ResNet-18. Since the results for these competitors in the tables are excerpted from EFC or LwF+LDC, the authors should align all performance results using the full ResNet-18 under equal conditions to accurately assess how much better the proposed approach is. This applies to both the stability gap plot and the per-task accuracy as well.

- In Figure 2 (top left), it is unclear why the accuracy on Task 1 for EFC before the training of Task 2 is significantly lower compared to LwF-SS and MITM, even though incremental learning has not yet been performed. This suggests that the starting points of the three approaches may differ, which could be related to the point I previously mentioned. Accuracy plots across the incremental learning steps would be necessary to clarify this. Additionally, the accuracy values in the stability plot are inconsistent when compared to Figure 3.

- The experimental section is limited. The authors only evaluate their approach on S-CIFAR100 (10 and 20 steps) and Tiny-ImageNet (10 steps), without conducting any evaluation on the ImageNet-Subset (a subset of ImageNet-1K, with the same resolution but only 100 classes). Additionally, a complete evaluation with 20 steps would be necessary (on both  Tiny-Imagenet and Imagenet-Subset). These benchmarks are standard and widely used [d][e][f][g], making them essential for understanding robustness of the method and comparisons in future works.

- The authors do not discuss the limitations of their approach. For instance, although the introduction of the middle model is interesting, it also introduces a computational training burden. While prototype-based approaches may be less effective, they do not significantly increase training resource requirements, as they only require training a single backbone per task. In contrast, the MITM approach requires training two backbones per task. How much does the computational burden increase compared to other approaches? Moreover, how does it compare to joint training on a single model?


[a] Lucas Caccia, Rahaf Aljundi, Nader Asadi, Tinne Tuytelaars, Joelle Pineau, and Eugene Belilovsky. New insights on reducing abrupt representation change in online continual learning. In ICLR, 2022

[b] Matteo Boschini, Lorenzo Bonicelli, Pietro Buzzega, Angelo Porrello, and Simone Calderara. Class-
incremental continual learning into the extended der-verse. TPAMI, 2022.

[c] David Lopez-Paz and Marc’Aurelio Ranzato. Gradient episodic memory for continual learning.
NeurIPS, 2017

[d] Gregoire Petit, Adrian Popescu, Hugo Schindler, David Picard, and Bertrand Delezoide. Fetril:
Feature translation for exemplar-free class-incremental learning. In WACV, 2023

[e] Simone Magistri, Tomaso Trinci, Albin Soutif, Joost van de Weijer, and Andrew D Bagdanov. Elas-
tic feature consolidation for cold start exemplar-free incremental learning. In ICLR, 2024

[f] Alex Gomez-Villa, Dipam Goswami, Kai Wang, Andrew D Bagdanov, Bartlomiej Twardowski, and
Joost van de Weijer. Exemplar-free continual representation learning via learnable drift compen-
sation. In ECCV 2024

[g] Fei Zhu, Xu-Yao Zhang, Chuang Wang, Fei Yin, and Cheng-Lin Liu. Prototype augmentation and
self-supervision for incremental learning. In CVPR, 2021

**Questions:**

From my perspective, the paper is well-written, and the methodology offers some novel contributions, making the proposed method appealing.

Regarding the methodology's weaknesses, while the introduction of the MITM model for class incremental learning is novel, further empirical analysis explaining why symmetric logit distillation acts as an effective regularizer to mitigate forgetting and reduce inter-task confusion [1] in the exemplar-free setting could enhance its novelty. In exemplar-based settings (e.g., X-DER) or exemplar-free settings with prototypes (e.g., LwF+LDC), the final classifier is trained to distinguish classes across tasks using exemplars or prototypes, making the mitigation of inter-task confusion in these scenarios more intuitive. Here, however, it is less clear how the proposed approach mitigates inter-task confusion, as the classifier is never explicitly trained to distinguish samples from different classes. Regarding representation forgetting [2], an analysis using linear probing could help to assess the separability of features, particularly in comparison with LwF+LDC, thereby providing insights into representation forgetting.

Finally, a clearer discussion on batch normalization and what is its impact in the improvement is required.

As for the experimental weaknesses, I have major concerns. The experimental section needs clarification, as I mentioned earlier. It is unclear whether all comparisons employ the same architecture for a fair evaluation. Furthermore, only a few experiments on multiple benchmarks are performed. While I am not asking for a large-scale evaluation on ImageNet-1K, a limited set of benchmarks with reasonable computational requirements is necessary for understanding robustness of the method and future comparisons. Finally, the authors should discuss the limitations of their approach, as providing a thorough overview is equally important.

Overall, given the above considerations, I rate the paper as marginally below the acceptance threshold,  leaning towards rejection. In its current state, I do not consider it ready for acceptance. I am open to adjusting my score, either increasing or decreasing it, based on the rebuttal phase.

Additional Questions (not relevant to the score): Is the leading model trained from scratch for each task, or is it a copy of the trailing model prior to the current task training? This detail seems to be missing from the paper and could be worth including

[1] Marc Masana, Xialei Liu, Bartlomiej Twardowski, Mikel Menta, Andrew D. Bagdanov, Joost van de Weije, Class-incremental learning: survey and performance evaluation (TPAMI 2022)

[2] MohammadReza Davari, Nader Asadi, Sudhir Mudur, Rahaf Aljundi, Eugene Belilovsky, Probing Representation Forgetting in Supervised and Unsupervised Continual Learning (CVPR 2022)

---

### Official Review · Reviewer_R7Zj · 2024-10-29

**Soundness:** 2
**Presentation:** 3
**Contribution:** 2
**Rating:** 5
**Confidence:** 5

**Summary:**

This paper proposes an exemplar-free continual representation learning method with Symmetric Distillation to address catastrophic forgetting in continual learning. The method introduces a Model-in-the-Middle (MITM) architecture that divides the network into leading, middle, and trailing models. The leading model focuses on learning new tasks, the trailing model preserves knowledge from previous tasks, and the middle model distills knowledge from both. This design aims to balance stability and plasticity effectively, reducing task-recency bias. Experimental results show that this approach outperforms existing methods on several benchmarks in exemplar-free, cold-start continual learning settings.

**Strengths:**

1.Divide the network to be optimized into leading, middle, and trailing versions, allowing the network to learn new knowledge while preserving previous knowledge, thus effectively balancing stability and plasticity
2.A new metric Final Accuracy standard deviation (FAσ) is introduced to measure the degree of drift in the model's representation ability for old tasks during continual learning.
3.The paper presents extensive experiments on multiple benchmark datasets, demonstrating the method's superiority, especially under the stringent exemplar-free, cold-start setting.

**Weaknesses:**

1.Related work section is overly broad and insufficiently detailed, and it does not provide a brief overview of the state-of-the-art methods used for comparison later, such as EFC [1].
2.It remains to be clarified what challenges are encountered by cutting-edge continual learning research in the exemplar-free, cold-start setting, and which directions current research is focused on.
3.The MITM method employs symmetric distillation; however, the loss function is calculated using logits without aligning feature extraction with the trailing model, which limits the interpretability of this approach. Moreover, as the number of tasks increases, this may lead to a drift in the model’s feature representation capability, potentially explaining the performance decline in the exemplar-based setting.
4.There is a lack of discussion on parameter count and computational overhead. The MITM architecture requires maintaining three models simultaneously, which incurs additional computational and memory burden. This paper does not clarify the method’s performance in terms of computational efficiency and memory usage.

[1] Simone Magistri, Tomaso Trinci, Albin Soutif, Joost van de Weijer, and Andrew D Bagdanov. Elastic feature consolidation for cold start exemplar-free incremental learning. In ICLR, 2024.

**Questions:**

- In Equation 8, only one hyperparameter λ is introduced. Would it be possible to introduce another hyperparameter for L_trialing?
- Table 6 shows the performance of MITM in the exemplar-based cold-start setting, with an unexpected decline across various metrics. This result seems somewhat counterintuitive, and further experiments are needed to explain the underlying reasons.

---

### Official Review · Reviewer_r4ct · 2024-11-03

**Soundness:** 3
**Presentation:** 3
**Contribution:** 3
**Rating:** 5
**Confidence:** 5

**Summary:**

The paper propose a continual learning method which does not depend on prototypes, exemplars or pre-training stage. The authors propose to separately the representations of old and new tasks using different models and then propose to distill their knowledge to a middle model to ensure a better stability-plasticity tradeoff. The proposed method outperforms existing methods across 2 datasets on cold-start CL settings with a reduced task-recency bias.

**Strengths:**

1. The paper is well-written, organized and easy to understand.
2. The paper proposed a very simple training strategy of symmetric distillation with 3 models.
3. The Final Accuracy standard deviation metric is quite helpful for evaluation.
4. The discussion of stability gap is appreciated.

**Weaknesses:**

1. References and discussion in comparison to similar CL methods using auxiliary networks [1] or learning with multiple models [2] are missing.

2. Poor experimental section: It is a standard practice to evaluate the method on a large resolution dataset like Imagenet-100 as done in all recent works like [3,4], while this paper provide experiments only on small resolution datasets like CIFAR100 and TinyImageNet. Recently proposed method - Adversarial Drift Compensation [3] also used the challenging cold-start setting and should be included in the comparison.

3. It is not clear how the proposed method works in warm-start settings. While it’s good to evaluate in challenging cold start settings, does the method still work in warm-start settings?

4. While warm-start evaluation can be a bit biased due to large number of base classes, a more realistic setting is using Pre-trained ViT models following recent works [5]. It would be interesting to see how MITM works with pre-trained models.

[1] Kim, Sanghwan, et al. "Achieving a better stability-plasticity trade-off via auxiliary networks in continual learning." Proceedings of the IEEE/CVF Conference on Computer Vision and Pattern Recognition. 2023.

[2] Arani, Elahe, Fahad Sarfraz, and Bahram Zonooz. "Learning Fast, Learning Slow: A General Continual Learning Method based on Complementary Learning System." International Conference on Learning Representations, 2022.

[3] Goswami, Dipam, et al. "Resurrecting Old Classes with New Data for Exemplar-Free Continual Learning." Proceedings of the IEEE/CVF Conference on Computer Vision and Pattern Recognition. 2024.

[4] Magistri, Simone, et al. "Elastic Feature Consolidation For Cold Start Exemplar-Free Incremental Learning." The Twelfth International Conference on Learning Representations.

[5] G. Zhang, L. Wang, G. Kang, L. Chen, and Y. Wei. Slca: Slow learner with classifier alignment for continual learning on a pre-trained model. In ICCV, 2023.

**Questions:**

It is not clear if all the compared methods used the same network “slimmed ResNet18”. This should be clarified.

---

### Official Review · Reviewer_mrJX · 2024-11-03

**Soundness:** 2
**Presentation:** 3
**Contribution:** 1
**Rating:** 3
**Confidence:** 4

**Summary:**

This work presents a new method for exemplar-free class-incremental learning (EFCIL) called Model-in-the-Middle (MITM). The study focuses on the cold-start scenario, where the trade-off between plasticity and stability is more pronounced than in typical incremental learning settings. MITM proposes using symmetric distillation across three models: a leading (plastic) model, a trailing (stable) model, and a middle (student/learner) model. The authors demonstrate empirical results on equally split CIFAR-100 and Tiny-ImageNet datasets.

**Strengths:**

1. Tackling one of the hardest problem of CL - EFCIL in a cold-start scenario.
2. The paper is easy to follow, with a good explanation of the used evaluation metrics.

**Weaknesses:**

1. Such method of double distillation from fully plastic and second stable model is not knew and it was presented in previous works, i.e., [1], [2]. [1] presents far more results and evaluation on as well CIFAR-100 and TinyImageNet. But this work does not compare to it (even to LwF+ANCL).

The second work [2] extend it even a bit, proposing that the "copy" operation can be exchanged to distillation and stable model can be different than the plastic one (heterogenous architectures), however, here for the unsupervised continual representation learning.

2. Choice of FeTrIL for cold start scenario in EFCIL is questionable. Providing one result for DMC in Tab.1 for comparison as well.

3. Some not clear statements in the text, e.g. line 320-321: _our method optimizes all three models jointly on the current task data._ I believe that is not true, because the trailing model is frozen - see Fig. 1 (BN are only updated to update a teacher). This can be confusing for the reader.

4. Related work: positioning methods like LwF, EWC as a warm start methods. More focus on warm-start than cold-start related work, and selection of the methods.

5. Using only two, small-size datasets and a single model architecture.

Overall, I find this work a very simple research with an already existing idea, limited experimentation section and non-insightful analysis of the results (ok, the bias is presented nicely), with a strange selection of the method for comparison. I think that this work need a bit of improvement and more experiments for bringing something new and be a good publication.

[1] Kim, Sanghwan, et al. "Achieving a better stability-plasticity trade-off via auxiliary networks in continual learning." Proceedings of the IEEE/CVF Conference on Computer Vision and Pattern Recognition. 2023.
[2] Gomez-Villa, Alex, et al. "Plasticity-optimized complementary networks for unsupervised continual learning." Proceedings of the IEEE/CVF Winter Conference on Applications of Computer Vision. 2024.

**Questions:**

1. Why work do not present the results on the bigger datasets, e.g., ImageNet-100?
2. Why not using lambda=0.8 for the all results if that presents the better outcome for FAA, AIA, and Forgetting than 1.0?
3. Why in the abstract not state that is class-incremental learning?
4. Can you elaborate/support this claim? (l.61-62:) _Moreover, exemplar-based solutions do not generalize to vision-language domains, as keeping exemplars of all possible textual descriptions is not tractable_

---

### Author Response · Authors · 2024-11-13

We want to thank the reviewers for their time and valuable feedback on the paper. We appreciate the positive comments on the writing, results, and clarity of the method. While we believe that we can address all comments, this effort would extend beyond the timeframe of the rebuttal. We therefore will withdraw the paper and we will focus on improving the paper based on the comments from the reviewers for a future re-submission. Thank you again for your input, it will help us make a better version of the paper.

---

### Note · Authors · 2024-11-13

**Comment:**

We want to thank the reviewers for their time and valuable feedback on the paper. We appreciate the positive comments on the writing, results, and clarity of the method. While we believe that we can address all comments, this effort would extend beyond the timeframe of the rebuttal. We therefore will withdraw the paper and we will focus on improving the paper based on the comments from the reviewers for a future re-submission. Thank you again for your input, it will help us make a better version of the paper.

**Withdrawal Confirmation:**

I have read and agree with the venue's withdrawal policy on behalf of myself and my co-authors.